# An Arabic Sedentary Behaviors Questionnaire (ASBQ): Development, Content Validation, and Pre-Testing Findings

**DOI:** 10.3390/bs12060183

**Published:** 2022-06-08

**Authors:** Hazzaa M. Al-Hazzaa, Shaima A. Alothman, Nada M. Albawardi, Abdullah F. Alghannam, Alaa A. Almasud

**Affiliations:** Lifestyle and Health Research Center, Health Sciences Research Center, Princess Nourah Bint Abdulrahman University, Riyadh 11671, Saudi Arabia; shaalothman@pnu.edu.sa (S.A.A.); namoalbawardi@pnu.edu.sa (N.M.A.); afalghannam@pnu.edu.sa (A.F.A.); aaalmasud@pnu.edu.sa (A.A.A.)

**Keywords:** Arab, health behavior, questionnaire, sedentary behavior, self-report, sitting time, validity

## Abstract

Background: Sedentary behaviors (SB) are very prevalent nowadays. Prolonged sitting associates with chronic disease risks and increased mortality even while controlling for physical activity. Objective measurement of SB is costly, requires technical expertise, and is challenging in terms of time and management. Currently, there is no validated self-reported instrument in the Arabic language that assesses SB among individuals and relates sedentary time to social, environmental, and health outcomes. The aim of this research was to develop a multi-item Arabic SB questionnaire (ASBQ). Methods: The ASBQ was developed through an extensive literature review and discussion by the research team (*n* = 5), then went through content validation (*n* = 10 experts) and pre-testing using cognitive interviewing procedures (*n* = 51 respondents, mean (SD) age was 38.3 (18.2) years, and with 49% females). Results: The ASBQ included 13 questions comprising a wide range of sedentary activities. The Arabic SB instrument showed excellent content validity for assessing sedentary time in adolescents and adults with a very high item-level and scale-level content validity index. A kappa statistic, a measure of interrater reliability, was 0.95. The pre-testing showed that the instrument was highly rated by a diverse sample of Saudi adolescents and adults. Conclusion: The ASBQ received excellent acceptance by a panel of experts with promising pre-test results. Further testing of psychometric properties, including test-retest reliability and criterion validity is required.

## 1. Background

Sedentary behavior (SB) has recently been defined as “any waking behavior characterized by an energy expenditure ≤1.5 metabolic equivalents (METs) while in a sitting or reclining posture”, excluding sleep [1]. SB is a noticeably common behavior. Serial testing of a large US population showed that the prevalence of sitting while watching television or videos at least two hours per day was high (ranging from 59% to 65%), and the estimated total sitting time increased from 2007 to 2016 from 7.0 to 8.2 h per day among adolescents and from 5.5 to 6.4 h per day among adults [2]. The prevalence of SB (≥4.5 h of sitting time) among European adolescents was also found to be high, reaching 76.8% in 2017, with no gender difference [3]. A previous local study conducted on Saudi youth revealed that very high proportions of males (84%) and females (91.2%) were spending more than two hours per day on screen time [4]. In addition, a recent national survey conducted in 2019 (with a single-item question on sitting time and including over 9000 households from 13 administrative regions) showed that the mean sitting or reclining time among the Saudi population was 9.56 h per day [5].

SB is considered an important risk factor for mortality and several chronic diseases, including type 2 diabetes and cardiovascular disease, independent of moderate-to-vigorous physical activity levels [6,7,8,9]. Prolonged sitting has been linked to chronic diseases and increased risk of mortality even when controlling for leisure-time physical activity [10]. Epidemiological evidence indicates that sitting is associated with cardiovascular disease biomarkers, such as total cholesterol, LDL cholesterol, triglycerides, waist circumference, and blood pressure levels [11,12]. In a recent prospective study, a long-term reduction in sedentary behavior improved peripheral vascular function and cerebral blood flow in individuals with increased cardiovascular risk [13]. A longitudinal Danish study indicated that after adjusting for physical activity levels, total sitting time remained a risk factor for diabetes only in inactive and obese populations [14]. All this has recently led the World Health Organization to issue recommendations to reduce SB at all ages as part of their physical activity and SB guidelines [15]. However, one recent study concluded that “not moving (physical inactivity) is bad for you, but how you stay still probably doesn’t matter” [16].

Clear guidelines on sedentary sitting time for children do exist. It is recommended that children and youth should not spend more than two hours in recreational screen time daily [15]. A separate guidance on sedentary behavior in adults, however, was less clear, and most recommendations are embedded implicitly in physical activity guidelines [15]. In addition, breaking sitting time with physical activity has been regarded as an important health issue, as it was shown to be associated with positive cardiovascular risk markers, including lower waist circumference, triglycerides, and 2 h plasma glucose [17]. Research indicates that high levels of moderately intense physical activity (about 60–75 min per day) appear to eliminate the increased risk of death associated with high sitting time [18]. In another study, sitting was associated with all-cause and cardiovascular disease mortality risk among the least active participants and that an amount of moderate-to-vigorous physical activity equivalent to meeting the current recommendations reduced or effectively eliminated such associations [19]. Furthermore, when young non-obese men participated in two randomized three-hour sitting trials, there was significant impairment of shear rate, superficial femoral artery endothelial function, and flow-mediated dilation (FMD); however, introduction of light activity breaks after each hour of sitting prevented the decline in FMD [20].

SB can be assessed using objective or subjective methods [21]. The objective measurement of SB using accelerometers or/and inclinometers is found to be very useful in assessing time spent in sedentary activities [22]. However, such assessment cannot differentiate the specific mode of sedentary behaviors, such as viewing TV, reading, using computers, or playing electronic games while sitting. In addition, accelerometers must be worn continuously for several days, which may increase the attrition rate of study participants [23]. Thus, the use of questionnaires has been one of the most frequent approaches to assess and quantify SB [21]. Self-reported SB measures assess various modes of SB, provide valuable measures of risk associated with health indicators, and are more strongly associated with health outcomes than objective measures, especially among children and youth [7,21,24,25]. The current evidence [7] indicates that screen time (especially TV viewing) obtained by SB questionnaires has a bigger impact on health compared with overall sedentary time that is attained by motion sensors. In a recent systematic review, single-item measures performed more poorly than multi-item questionnaires [26]. Most importantly, due to their low cost and ease of implementation, questionnaires remain a method of choice for population-based large-scale epidemiological studies. To improve knowledge and utility of SB questionnaires and to reduce the gaps in assessing sedentary behaviors subjectively and their effects on health outcomes, it is essential that multiple characteristics of SB be assessed such as doing arts, crafts, and hobbies [27], while keeping the questionnaire as short and manageable as possible.

Currently, there is no validated self-reporting multidimensional instrument in Arabic that can be used to assess SB among adolescents or adults and relate such sedentary activity to health outcomes. A recent analysis of physical activity and SB (based on single-item questionnaires) in the Middle East and North Africa region suggests a significant prevalence of insufficient physical activity, increased SB among adults over the last two decades, and inconsistency in SB measurement due to the absence of standardized guidelines for quantification and interpretation [5,27,28]. Therefore, the purpose of the current research was to develop a culturally suitable multi-item Arabic SB questionnaire (ASBQ) to assess sedentary activities among a wide range of age groups in the Arab culture, along with its psychometric properties, including content, construct, and criterion validity, as well as internal consistency and reliability. However, this article presents the findings related to the development, content validity, and pre-testing of the ASBQ.

## 2. Methods and Procedures

### 2.1. Institutional Review Board (IRB) Approval and Parental Consent

The research project was approved by the Institutional Review Board at Princess Nourah Bint Abdulrahman University (IRB Log number: 20-0229). In addition, informed consent was obtained from all participants in the pre-testing phase. Participants below age 18 gave their own written consent and that of their parents. Confidentiality of data was ensured by coding and storage in restricted access files. Participants were informed that they could withdraw at any time.

### 2.2. Arabic SB Questionnaire (ASBQ) Development

Many instruments are presently available for the measurement of SB in adult and pediatric populations. Few questionnaires, however, describe SB across all common domains of daily life in the general population. The majority focus on a defined domain such as leisure time, workplace, or a specific population [29,30,31,32,33]. A recent review identified 35 adult questionnaires that have undergone psychometric testing [21]. Figure 1 shows the flow diagram of the research protocol for developing and testing the Arabic SB questionnaire. We first conducted an extensive literature search using PubMed and Google Scholar databases to identify existing SB questionnaires in the English language. We used terms such as “sedentary behaviors questionnaire”, “sitting”, “sedentary behaviors assessment”, “sedentary time questionnaire”, and “subjective assessment of sedentary activity”. After excluding duplicates and screening the titles and abstracts, the search resulted in 22 relevant SB questionnaires (detailed description is shown in Appendix A) that assessed sitting time via a multiple-item questionnaire [29,30,31,32,33,34,35,36,37,38,39,40,41,42,43,44,45,46,47,48,49,50]. Some studies were validated with objective measurements, but five were found to be more specific for assessing sedentary behaviors and were therefore selected and fully reviewed. Of these, only three questionnaires (two for adults and one for adolescents) were found suitable for our intent and purpose (that assesses SB and not just inactivity, includes all domains of SB and not only one domain, has reliability and validity data against objective measures, and was not very long) and were thus considered for further examination in the development of our Arabic sedentary behaviors questionnaire [33,35,47]. The latter instrument showed test-retest reliability among adult questionnaires of fair to good reliability (intra-class correlation coefficient (ICC) = 0.53–0.68), and for criterion validity, a significant correlation (0.52) was found for total sedentary time [33]. For adolescent questionnaires, the reliability showed moderate-to-substantial agreement for most (91%) items (ICC = 0.41–0.79), with three items (4%) reaching high agreement levels (ICC = 0.82–0.83) [35]. The third one, though has some good items that can be adopted, it also included other items on activity times [47].

After extensive discussion among research team members, it was noted that one of the SB questionnaires was very long, rendering it less practical. Also, some items in these questionnaires needed restructuring and culturally adapting. The scoring scales in most questionnaires may also need expansion to accommodate a wider range of sedentary times [27,28]. In addition, none of these questionnaires had a combined total sedentary time in hours and minutes per day. The research team, therefore, decided to develop an Arabic sedentary questionnaire, taking into account that the newly developed sedentary behaviors instrument needs to satisfy the following criteria:It should be practical when administered and take a reasonably short time to complete (maximum 10–15 min).Questions should cover all domains of SB activities that expend no more than 1.5 METs (sitting, reclining, and lying down while reading, viewing a screen and non-screen-based sedentary time, socializing, sitting in a car, bus, or train, sitting at school or at work, doing hobbies or household activities while seated, etc.).Can be applicable to a wide range of age groups from adolescents to elderly.Sedentary time is summed to account for total sedentary time in hours and minutes per weekday and on weekends.All possible psychometric properties testing must be conducted when developing and validating the instrument, including content validity, internal consistency, construct validity, reliability, and criterion validity, using an objective method such as the activPAL accelerometer as a criterion measure.

After adopting the commonly accepted SB construct [1], formulating the questionnaire went through several phases. In the early phase, we generated the questions, the scoring system, and the format. The questionnaire was then revised several times by the research team to ensure that its items were related to the concepts being investigated. The team was composed of five members with expertise in sedentary behaviors, physical activity, and lifestyle and health. They were all bilingual (English and Arabic). Extensive discussion was held, taking into account that the questionnaire must be comprehensive, specific, and at the same time not too long. General structure, question flow, clarity of instructional statements, and readability of the questions were considered on several occasions prior to content analysis and pre-testing. The second phase included content validity by a group of experts, and the research team then conducted a pre-testing phase. The target population for the ASBQ included individuals from age 12 years and above (12–65+ years), as research has shown that parental-reported measures of physical activity and SB were not useful as a proxy for 2–9 year-old children’s physical activity and sedentary time [51].

### 2.3. Content Validity Testing

Validity is defined as the ability of an instrument to measure the properties of the construct under investigation [52]. It addresses the degree to which an instrument’s items adequately represent the content domain. Content validity is considered a prerequisite for construct and criterion-related validity and should receive the highest priority during instrument development [53]. To test the content validity of the SB instrument, an expert panel of 12 bilingual individuals was appointed. From previous studies, the recommended number of experts for content validation appears to be between six and ten people [53,54]. The experts were from a broad spectrum of disciplines, including exercise physiology, physiotherapy, public health, physical activity, sports sciences, and behavioral science. All the experts held PhD degrees, were bilingual, and had sufficient experience in the field of physical activity and sedentary behaviors. It is recognized that as the number of experts increases, the probability of chance agreement decreases [55]. Ten of the twelve experts contacted completed the evaluation forms. The reason given for non-completion by the two invited experts was lack of time.

The non-face-to-face approach was used to conduct content validity. In a cover letter, clear instructions were provided to panel members to rate each of the instrument’s items in terms of clarity and relevance to the underlying construct, using a 4-point ordinal scale. For clarity, we used the following scoring system: 1 = unclear, 2 = needs major revision, 3 = needs minor revision, and 4 = very clear. Scoring the items’ relevance to sedentary behaviors included similar ranking: 1 = not relevant, 2 = needs major revision, 3 = needs minor revision, and 4 = highly relevant. A table was included with instructions to guide experts on the scoring method. The experts were requested to provide a score on each item independently based on the relevance scale. To analyze the content validity findings, we used the content validity index (CVI) by giving a rating of 4 or 3 as 1 point and ratings of 2 or 1 as zero [53,54,55,56].

### 2.4. Pre-Testing the ASBQ

The purpose of pre-testing is to improve the primary questionnaire and ultimately the response rate. It is intended to ensure that respondents can comprehend the questionnaire items and to determine whether rewording or restructuring is needed. It also reveals the ability of respondents to accurately recall and estimate time spent in each sedentary activity in providing their answers. The participants in the pre-testing represented a convenience sample recruited from Princess Nourah University, King Abdullah Bin Abdulaziz University Hospital, and from the Health Sciences Research Centers and nearby community. We excluded any person who had an illness or disorder that prevented him or her from moving freely (such as severe heart disease, respiratory disease or orthopedic problem). There were 51 participants (49% were females) between the ages of 12 and 80 years [57]. They came from a broad range of backgrounds and socioeconomic statuses, as having sufficiently diverse participants allowed exploration of as many aspects of the SB questionnaire as possible [58]. Collected data from the participants included age, sex, body weight and height, levels of education, working status (working, unemployed, retired), and comorbidity, such as obesity, diabetes, high blood pressure, and depression.

The ASBQ was administered in semi-structured cognitive interviews in pre-testing through face-to-face interviews (or by phone). Cognitive interviewing is an evidence-based qualitative method specifically intended to scrutinize whether a survey question fulfils its intended purpose [59]. Respondents were provided the questionnaire and asked to complete it without interruption or asking for clarification. Time spent answering the questions was recorded. Immediately after completing the sedentary behaviors questionnaire, a cognitive interview with verbal probing was conducted on relevance, importance, and whether some important domains or areas of sedentary activity were missing [58]. Examples of questions that were asked include the following:What do you think the question is about?Is the question clear and understandable? If not, how can it be made clearer?Do you have any questions about the items?How could the wording be clearer?Are there activities that we omitted?Did any of the questions make you feel uncomfortable?

### 2.5. Anthropometric Measurements

Body weight and height were reported by the participant to the nearest kg and cm, respectively, using calibrated medical scales (Seca medical scale, Hamburg, Germany). Body mass index (BMI) was calculated as the ratio of weight in kilograms by height in meters squared. BMI was used to differentiate between participants that were overweight/obese and those with normal weight.

### 2.6. Computational and Statistical Analyses of the Data

To obtain a content validity index for the relevancy of each item, the item-level content validity index (I-CVI) was calculated by averaging the ten experts’ ratings (scores of one or zero). Thus, I-CVI represents the number of experts giving a rating of 3 or 4 to the relevancy of each item divided by the total number of experts to express the proportion of agreement on the relevancy of each item [53,56]. If the I-CVI is above 79%, the item is appropriate. If it is between 70% and 79%, it needs revision. If it was less than 70%, it was eliminated [60]. We also calculated the scale-level content validity index (S-CVI). S-CVI represents the proportion of total items rated as content valid or those items achieving scores of 3 or 4 by the panel of experts [56]. In addition, the kappa statistic, an index of interrater agreement that also adjusts for chance agreement, was calculated as another indicator of content validity [55]. To calculate the kappa statistic, the probability of chance (Pc) agreement was first calculated for each item using the following formula: P_c_ = [N/A (N − A)] × 0.5^N^ (55). In this equation, N = the number of experts in a panel and A = the number of panelists who agree that the item is relevant. Finally, kappa was computed by entering the numerical values of the probability of chance agreement (P_c_) and the content validity index of each item (I-CVI) in the following equation: K = (I-CVI − P_c_)/(1 − P_c_). Appraisal criteria for kappa are that values above 0.74 are considered excellent, between 0.60 and 0.74 are considered good, and values between 0.40 and 0.59 are considered fair [61]. It is worth mentioning that as the number of experts in a panel increases, the probability of chance agreement diminishes and values of I-CVI and kappa converge [55]. We also calculated the content validity ratio (CVR) using the following formula: CVR = (Ne − N/2)/(N/2), in which Ne is the number of expert panelists indicating “essential” and N is the total number of panelists [60]. Data from the pre-testing phase were reported as means (SD) or frequencies and percentages. We also used Pearson’s or Spearman’s correlation coefficient to examine the relationships between selected variables.

For the total sedentary time, we capped the maximal possible time to 17 h per day, assuming a minimum sleep duration of seven hours per night. There were about 15% of the respondents who over-reported their total sedentary time per day. Confirmatory factor analysis was used to test the amount of variance shared among questionnaire items, using extraction method as principal components analysis and Varimax rotation with Kaiser normalization. Because our questions are repeated for weekdays and weekends, we separated the factors for the two components (weekdays and weekends). We used 13 items (questions) for each component after excluding questions 1, 2, and 16. The Kaiser-Meyer-Olkin (KMO) measure of sampling adequacy and Bartlett’s test of sphericity indicated a significant value for weekday items (KMO = 0.497, *p* = 0.019) but an insignificant value for weekend items (KMO = 0.550, *p* = 0.193). However, we wanted to obtain the total variance explained by the factors in the data matrix for each weekday and weekend item. We also reported the initial eigenvalues, the percent (%) of variance, and the percentage of cumulative variance in the rotation sums of squared loadings. For statistical analysis, we used IBM-SPSS software, version 22 (Chicago, IL, USA).

## 3. Results

The ASBQ items, translated to English, are shown in Table 1 (its complete form in Arabic language is presented in Appendix A and English translation is shown in Appendix A). The items cover a set of questions related to time spent sitting, reclining, and lying down in different contexts. The questionnaire items included questions prompting the respondent to report time spent being sedentary (sitting) in different settings during school, work, transportation, and leisure time. These included time spent on TV viewing, computer use at home or work, playing videogames while sitting, reading, sitting to chat with friends/relatives, listening to religious verses or music, speaking on the phone, resting (lying down but not napping), using transportation (car, bus, train, subway, or motorbike) while seated, doing crafts, hobbies, or art work (drawing, knitting, sewing, etc.) while seated, and doing household tasks (ironing, slicing food, repairing things, etc.) while seated. Respondents were cautioned to identify only the focus of sedentary activity during a given time period. For example, when reading and listening to music at the same time, only one activity should be counted. The instrument asks for the average time spent during a typical (usual) week during both school/work hours and out of school/work hours. In addition, the ASBQ included an item related to how often the respondent interrupts his/her daily sitting time.

### 3.1. Content Validity Findings

The majority of the experts (80%) assessed clarity by indicating that the items were mostly very clear, with few items judged as needing minor revision. None answered that an item was unclear. The research team went over the experts’ answers about clarity and revised the items accordingly.

Table 2 presents the relevance ratings of the ten experts on the item scale and the content validity index calculation. As seen in the table, average item relevance ranged from 0.88 to 1 (1 = perfect rating), with a mean item-level content validity index of 9.56 of 10. The mean scale-level content validity index was 0.96 (1 = perfect), whereas the mean modified kappa agreement was 0.96.

The relevance versus non-relevance scores and interpretation of the item-level content validity index are shown in Table 3. Based on the item-level content validity index, all questionnaire items except one showed an appropriate rating. Question 14 needed revision, which was performed after content analysis and throughout the results of pre-testing. Content validity ratio ranged from 0.40 to 1.0, with 12 items showing a perfect score.

### 3.2. Pre-Test Findings

As shown in Table 4, a total of 51 participants completed the ASBQ and participated in the cognitive interview. Their ages ranged from 12 to 80 years, and 25 of the 51 were female (49%). The mean (SD) of reported weight and height were 72.7 (15.3) kg and 162.4 (8.1) cm, respectively. Also, the mean value (SD) of BMI was 27.66 (5.9) kg/m^2^. About 45% (23 of 51) reported at least one comorbidity or risk factor, and 41.2% (21 of 51) were working in an office. The mean (SD) of the total time taken to complete the questionnaire was 11.1 (3.9) min.

In addition, 33.3% of the pre-testing sample were classified as obese. Nearly 55% of participants had a college degree or higher. Approximately 28% of the sample were not currently working (either unemployed, underage, or retired).

The results of the cognitive interview of the ASBQ are shown in Table 5. Overall, the vast majority of the participants understood the intended meaning of the questionnaire. Also, 95.6% of the respondents believed that the content was clear, while 96.7% of the sample thought the wording was clear to them.

Suggestions and feedback from the respondents included adding religious practice while seated (*n* = 5), creating separate questionnaires for workers, students, and retirees (*n* = 4), giving more examples to clarify questions (*n* = 4), using an electronic version (*n* = 3), adding meal time (*n* = 3), adding sleep time (*n* = 1), creating separate versions for males and females (*n* = 1), providing pre-fixed times such as one hour, two hours, etc. (*n* = 1), adding waiting time for doctor visits (*n* = 1), and adding time sitting in a movie theater (*n* = 1).

Means (SD) and correlation coefficients between participants’ characteristics and total sedentary time during weekdays and weekends are shown in Table 6. There were no significant differences between the selected variables relative to total sedentary time on both weekdays and weekends. As for the correlational analysis, the only significant sedentary time was across BMI categories, as underweight participants spent significantly more sedentary time than overweight or obese participants.

Table 7 displays the correlation coefficients of total sedentary time with the questionnaire items for weekdays and weekends. Total sedentary time during weekdays was significantly correlated with items 3 (*p* < 0.001), 4 (*p* < 0.001), 8 (*p* = 0.001), 9 (*p* = 0.011), and 13 (*p* = 0.037). On the other hand, sedentary time during weekends correlated significantly with items 3 (*p* < 0.001), 4 (*p* = 0.001), 7 (*p* = 0.015), and 9 (*p* = 0.007).

The percent variances explained by the components of the questionnaire for weekday and weekend items are shown in Table 8. Based on a minimal Scree value of 1.0, six components of weekday items and five of weekend items have total eigenvalues above 1.0. The cumulative percent variance that can be explained by the weekday components was 70.14%, whereas the five weekend components with total eigenvalues above 1.0 can explain 61.59%. Moreover, the results of factor analysis with the rotated component matrix are shown in Table 9. They reveal the existence of six factors (dimensions) for the sedentary behavior items on weekdays and five factors for the weekends. For example, component one included doing simple crafts or artwork while sitting (item 10), reading for fun (item 5), listening to Quran, radio, or music (item 7), and engaged in other sitting activities (item 15), with coefficients ranging from 0.814 to 0.499, while component six included doing homework/studying (item 4) with a coefficient of 0.906 (see Appendix A for the complete items description).

## 4. Discussion

The purpose of the current study was to develop a comprehensive questionnaire that is culturally suitable for assessing a variety of sedentary activities in the Arab population along with its psychometric properties. The Arabic SB questionnaire was developed through extensive literature review and discussion by the research team (*n* = 5), and then went through content validation (*n* = 10) and pre-testing accompanied by cognitive interviewing procedures (*n* = 51). It comprised 16 questions, in which 13 items involved a variety of sedentary activities covering leisure-time activities, work, transport, household seated activities, reading, chatting, and listening to religious verses or music. ln addition, questions one and two deal with the number of days and hours the participant goes to work or school and there are instructions to skip these questions if the respondent is retired or unemployed. Question 16 deals with times the respondent interrupts sitting time (see Appendix A for full description of the instrument). The main findings showed that the Arabic SB questionnaire has excellent content validity for assessing sedentary time in adolescents and adults with a very high item-level and scale-level content validity index as well as a kappa statistic of 0.96. Further, the pre-testing phase showed that the instrument was highly rated and well received by a diverse sample of Saudi adolescents and adults. In general, the developed ASBQ has several features. It covers many domains of SB and included sedentary activities that are not available in some of previous questionnaires, such as doing arts, crafts, hobbies, reading, chatting, and listening to religious verses (particularly in Arabic) or music, while at the same time keeping the questionnaire as short and manageable as possible. It assesses both weekday and weekend sedentary time. It is applicable to a wide range of ages and has sitting data in hours and minutes for individual items and total time.

Content validity results of the present ASBQ showed a high item-level content validity index (I-CVI) averaging 0.96. This index represents an average expert rating for each item, and any I-CVI greater than 0.78 would fall into the range considered excellent, regardless of the number of experts [59]. Moreover, with an excellent kappa statistic of 0.96, the Arabic SB questionnaire reflected strong agreement among the panel of experts [61]. The process of pre-testing enabled us to refine the questions to reach their current form. The present findings showed that the lowest eigenvalue reported in our data was 0.261. Eigenvalues represent the total amount of variance that can be explained by a given principal component. In theory, they can be positive or negative; however, in practice, they explain variance that is always positive. As long as the eigenvalue is above zero, it is considered a good sign [62]. Also, our factor analysis findings show no item multicollinearity. In addition, the chi-square test appears to be the most commonly used fit index in confirmatory factor analysis [62]. Our model was evaluated for goodness of fit using the chi-square formula. However, knowing that the chi-square fit statistic is influenced by large samples, the ratio of the chi-square statistic to the respective degrees of freedom (χ^2^/df) was used [62]. It was suggested that a ratio of ≤2 indicates a superior fit [63]. In the present analysis, the chi square to degree of freedom ratios were 1.36 and 1.14 for items in weekday and weekend questions, respectively.

According to a recent study, sedentary behavior questions must be concise, valid/reliable, evidence-based, and developed using best practices [64]. Our SB questionnaire is composed of 16 items that can be answered within 10–15 min. Therefore, it can be conveniently incorporated into public health surveys. Indeed, comprehensive and inclusive local population surveillance will be enhanced by including the current SB instrument, especially considering that Saudi Arabia’s national health strategy has identified applied research on behavioral risk factors for non-communicable diseases as one of the key strategic approaches to be undertaken by the country [65]. Also, the recent Saudi Vision 2030 has stressed the importance of a healthy lifestyle in improving the health and prosperity of all segments of the Saudi population [66]. The current instrument will hopefully facilitate measurement of sedentary time, which is considered an indicator of unhealthy lifestyle behaviors.

The field of sedentary behaviors is relatively young; only in the last two decades has it received increased attention in research [1,64]. A survey intended to assess SB has the potential to reach a more demographically and geographically diverse group of participants than could be achieved with objectively assessed collection methods. In addition, objective measurement of sedentary behaviors may increase attrition rates [23]. Above all, due to their low cost and ease of implementation, questionnaires remain a method of choice for population-based surveying. However, a systematic review and meta-analysis study showed that logs and diaries are recommended to assess self-reported SB due to their higher validity and reliability than data collected by questionnaires [67]. Nonetheless, due to time and resource constraints, a sedentary behaviors questionnaire with minimal items may be preferable to other types of subjective assessment of SB in a large-scale population survey, particularly when its validity and reliability is showing similarity to those of longer questionnaires.

Cognitive interviewing is an evidence-based qualitative method specifically intended to scrutinize whether a survey question fulfils its intended purpose [59]. It is widely acknowledged that data obtained from a measure are only as valid as the items included in that measure [59]. Therefore, it is critical that items be rigorously constructed through good practices in question formulation and are subject to review and modification before reaching the pretesting stage, which was achieved in the present questionnaire. Additionally, in the present study, cognitive interviews were conducted by a group of researchers with similar expertise, which enhanced the quality of the cognitive interviews. Cognitive interviewing normally includes a small sample that may involve just 10 to 30 participants in total [59]. However, since our research project targeted a large age-range, from adolescents to retirees, we recruited more participants to better represent all age categories.

Our SB questionnaire included the same questions for both weekdays and weekend days. This allowed catching differences in sedentary behaviors between time at school or work and time out of school or off work. Such a format has been recommended previously, as most individuals have different lifestyle habits on weekdays versus weekend days [21]. Furthermore, people are increasingly engaged in simultaneous multi-tasking sedentary activities [21]. They may read or play games and at the same time listen to music. Therefore, summing such activities will inflate the amount of sedentary time, as earlier research has shown [68]. Therefore, in our questionnaire, we alerted respondents to this point in the instructions preceding the questions. Such an approach is recommended to avoid inflating the assessed sedentary time [21]. In addition, the Arabic SB questionnaire included an item relating to how often respondents interrupt their daily sitting time. Research has shown that increased breaks during sedentary time are beneficially associated with waist circumference, independent of total sedentary time and time expended in moderate-to-vigorous intensity activities [17].

Having an Arabic SB questionnaire to assess sedentary time is very important for research linked to chronic diseases and lifestyle behaviors in this region where physical inactivity and sedentary behaviors are quite prevalent [5,28,69]. The concept of sedentary behaviors is now recognized as a different entity from physical inactivity [1], and sedentary behaviors are known to be independently associated with risk factors for chronic disease and mortality [6,70]. In fact, physical inactivity and sedentary behaviors are believed to be associated with different adverse health outcomes [71]. Excessive screen viewing time among adolescents appears to be related to cardiovascular disease risks [72]. In addition, physical activity among preadolescents appears to be more closely associated with healthy food choices, whereas sedentary behaviors seem to relate largely to unhealthy dietary choices [73]. Furthermore, findings from lifestyle research conducted on Saudi youth have indicated that physical activity and sedentary behaviors are associated with different dietary behaviors. Healthful dietary habits (intakes of breakfast, fruit, vegetables, and milk/dairy products) are associated mostly with increased levels of physical activity, whereas unhealthful dietary habits (higher consumption of sugar-sweetened drinks, fast foods, cake/donuts, and energy drinks) are related to higher screen time [74]. Therefore, assessing SB as a separate risk factor from physical inactivity is an important public health issue.

These findings highlight the significant implications of the development of the current ASBQ on the future assessment of sedentary behaviors among the people in this region. Increased sedentary time is a public health challenge facing Saudi Arabia and the entire Arab region [28]. It is anticipated that the SB instrument developed and validated in this study will be well received by local and regional researchers and will hopefully contribute to greater understanding of the sedentary behavior of Arab people in general and the Saudi population in particular, including its determinants and association with social, environmental, and health outcomes. After greater reliability and criterion validity measures yet to be conducted for this SB instrument, it will be more useful and can play a significant role in testing future interventions aimed at reducing sedentary behaviors among the Arab population.

### 4.1. Strengths and Limitations

Certain strengths and limitations of this study can be mentioned. One strength is that we used cognitive interviewing to test the entire survey. The heterogeneity of the sample (adolescents, young adults, middle-aged, and older adults) used in the pre-testing can be identified as a key strength of this study. In our pre-testing, we used a retrospective probing method, as opposed to concurrent probing. Concurrent probing can disrupt the interview and lengthen the answers and probing process [58]. The limitations of the study that should be noted include, first, that although the questionnaire refers to a habitual (usual) week, potential recall bias is an inherent issue with subjective assessment methods. Second, experts’ feedback is considered subjective in any case, which may introduce bias. However, expert opinions are important as they improve the structure and enhance rephrasing of some wording. They may also add new items or content to the instrument [75]. Third, the current sedentary behavior questionnaire did not account for sitting at mealtimes. The reason is that mealtimes are usually standard and cannot be eliminated, as the quality of mealtimes are important for proper and adequate nutrition. Fourth, the instrument is not intended to be used with children under the age of 12 years, due to their inability to recall the details of the questionnaire.

### 4.2. Future Steps

The ASBQ will undergo further testing of psychometric properties, including internal consistency, construct validity, test-retest reliability, and criterion validity against an objective measure of SB, such as the activPAL activity/sedentary behaviors monitor and recorder. Indeed, testing has already begun and will continue for the next several months.

## 5. Conclusions

The ASBQ was developed through a preliminary literature review and discussions by the research team (*n* = 5), and then underwent content validation (*n* = 10 experts) and pre-testing using cognitive interviewing procedures (*n* = 51 participants). It included a total of 16 questions providing overall as well as specific SB estimates during a variety of sedentary activities while covering leisure-time activities, work, transport, household seated activities, reading, chatting, and listening to religious verses or music. The main findings showed that the ASBQ has high content validity for assessing sedentary time in adolescents and adults with an excellent item-level and scale-level content validity index and a kappa statistic of 0.96. Further, the pre-testing phase showed that the instrument was highly rated and well received by a diverse sample of Saudi adolescents and adults. The results of the factor analysis reveal the existence of six factors (dimensions) for sedentary behavior items for weekdays and five for weekends. The cumulative percent variance that can be explained by the weekday components was 70.14%. Further testing of psychometric properties, including test-retest reliability and criterion validity against an objective measure of SB is required.

## Figures and Tables

**Figure 1 behavsci-12-00183-f001:**
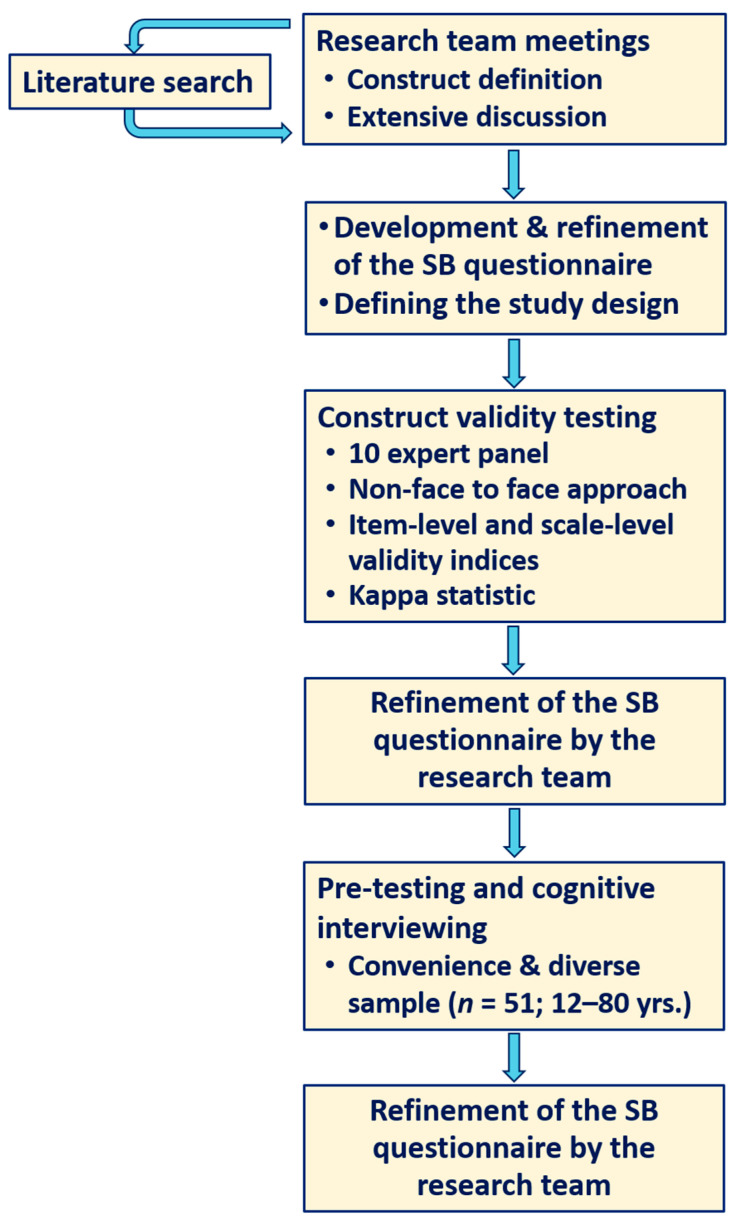
Flow diagram of research protocol for developing and testing the Arabic sedentary behaviors (SB) questionnaire.

**Table 1 behavsci-12-00183-t001:** The translated Sedentary Behavior Questionnaire (SBQ) showing the domains and the specific items. The question was asked as “how many hours and min per day you spend sitting, lying or reclining while during a typical day?”.

Domain	Item	Time
Leisure sitting time	1. Watching movies, TV, videos (regardless of the source-TV, computer, phone)	[ ] hrs. [ ] min
	2. Playing computer/video games	[ ] hrs. [ ] min
	3. Internet surfing or using social media for fun	[ ] hrs. [ ] min
Education or mentally active reading sitting time	4. Doing homework/studying	[ ] hrs. [ ] min
	5. Reading for fun	[ ] hrs. [ ] min
Talking, listening or resting sitting time	6. Sitting and talking with family or friends in person or via internet	[ ] hrs. [ ] min
	7. Listening to Quran, Radio, or music (without doing anything else)	[ ] hrs. [ ] min
	8. Resting (lying down, but not taking a nap, etc.)	[ ] hrs. [ ] min
Doing craft or hobby sitting time	9. Doing hobbies that require thinking/reasoning (doing puzzles, playing cards, doing crossword puzzles, etc.)	[ ] hrs. [ ] min
	10. Doing simple crafts or art work while sitting (like drawing, painting, knitting, sewing, etc.)	[ ] hrs. [ ] min
Transportation sitting time	11. Using transportation while sitting (in car, bus, train, subway or motorbike)	[ ] hrs. [ ] min
Household sitting time	12. Doing household tasks while seated (cooking, ironing, slicing foods, etc.)	[ ] hrs. [ ] min
	13. Engaged in other activities than the above while seated, name them:	[ ] hrs. [ ] min

**Table 2 behavsci-12-00183-t002:** The relevance ratings on the item scale by ten experts and content validity index calculation.

Item No.	Experts Relevance Ratings										Expert Agreement	Item-Level Content Validity Index (I-CVI)	Universal Agreement (UA)	Modified Kappa Agreement
	1	2	3	4	5	6	7	8	9	10				
Q-1	1	1	1	1	1	1	1	1	1	1	10	1	1	1
Q-2	1	1	1	1	1	1	1	1	1	1	10	1	1	1
Q-3	1	1	1	1	1	1	1	1	1	1	10	1	1	1
Q-4	1	1	1	1	1	1	1	1	1	0	9	0.90	0	0.90
Q-5	1	1	1	1	1	1	1	1	1	1	10	1	1	1
Q-6	1	1	1	1	1	1	1	1	1	1	10	1	1	1
Q-7	1	1	1	1	1	1	1	1	1	1	10	1	1	1
Q-8	1	1	1	1	1	1	1	1	1	1	10	1	1	1
Q-9	1	1	1	1	1	1	1	1	1	1	10	1	1	1
Q-10	1	1	1	1	1	1	1	1	1	1	10	1	1	1
Q-11	1	1	1	1	1	1	1	1	1	1	10	1	1	1
Q-12	1	0	1	1	1	1	1	1	1	1	9	0.90	0	0.90
Q-13	1	1	1	1	1	1	1	1	1	1	10	1	1	1
Q-14	0	1	1	0	1	0	1	1	1	1	7	0.70	0	0.69
Q-15	1	1	1	1	1	1	1	1	1	1	10	1	1	1
Q-16	1	1	1	0	1	1	1	1	1	0	8	0.80	0	0.80
Average Item relevance	0.94	0.94	1	0.88	1	0.94	1	1	1	0.88	9.56	S-CVI = 0.96	S-CVI/UA = 0.75	0.96
Average relevance across the 10 experts = 0.96														

Q = question. I-CVI = Item-level content validity index = average of expert rating for each item [53,56]. Universal Agreement (UA): rating with all 1 = 1, and any rating with 0 = 0. S-CVI = scale-level content validity index = average of I-CVI. S-CVI/UA = the average of universal agreement scores across all items. P_c_ = probability of chance agreement (PC = [N/A (N − A)]* 0.5^N^), where N = number of experts in a panel and A = number of panelists who agree that the item is relevant [53,56]. Modified Kappa agreement (K) = (I-CVI − P_c_)/(1 − P_c_) [55].

**Table 3 behavsci-12-00183-t003:** The relevance versus non-relevance scores and interpretation of the item-level content validity index.

Item No.	Relevant (Rating 3 or 4)	Not Relevant (Rating 1 or 2)	Item-Level Content Validity Index (I-CVI) *	Content Validity Ration **	Interpretation
Q-1	10	0	1	1	Appropriate
Q-2	10	0	1	1	Appropriate
Q-3	10	0	1	1	Appropriate
Q-4	9	1	0.90	0.80	Appropriate
Q-5	10	0	1	1	Appropriate
Q-6	10	0	1	1	Appropriate
Q-7	10	0	1	1	Appropriate
Q-8	10	0	1	1	Appropriate
Q-9	10	0	1	1	Appropriate
Q-10	10	0	1	1	Appropriate
Q-11	10	0	1	1	Appropriate
Q-12	9	1	0.90	0.80	Appropriate
Q-13	10	0	1	1	Appropriate
Q-14	7	3	0.70	0.40	Need Revision ***
Q-15	10	0	1	1	Appropriate
Q-16	8	2	0.80	0.60	Appropriate

* Interpretation of I-CVIs: If the I-CVI is higher than 79 percent, the item is appropriate. If it is between 70 and 79 percent, it needs revision. If it is less than 70 percent, it is eliminated [60]. ** The formula of content validity ratio is CVR = (Ne − N/2)/(N/2), in which the Ne is the number of expert panelists indicating “essential” and N is the total number of panelists. [60]. *** Slightly revised after content analysis and pre-testing.

**Table 4 behavsci-12-00183-t004:** Descriptive characteristics of the pre-tested participants (*n* = 51).

Item	Category	Value
Age (years)	Mean (SD)	38.3 (18.2)
Age category	Adolescents—12–17 years (%)	23.5% (12)
	Young Adults—18–35 (%)	23.5% (12)
	Middle age—36–49 (%)	23.5% (12)
	Older Adults 50+ (%)	29.5% (15)
Sex (%)	h	49% (25)
Body weight (kg)	Mean (SD)	72.9 (15.3)
Height (cm)	Mean (SD)	162.4 (8.1)
BMI (kg/m^2^)	Mean (SD)	27.7 (5.9)
	Underweight (%)	5.9% (3)
	Normal weight (%)	29.4% (15)
	Overweight (%)	31.4% (16)
	Obese (%)	33.3% (17)
Education	High school or less (%)	45% (23)
	College degree (%)	33.4% (17)
	Postgraduate degree (%)	21.6% (11)
Working status	Not working (%)	27.4% (14)
	Working online (%)	31.4% (16)
	Working in-person (%)	41.2% (21)

**Table 5 behavsci-12-00183-t005:** Results of cognitive interview of the Arabic SB questionnaire (*n* = 51).

Item Number (Question)	Participant Understanding of the Intended Meaning	The Content Was Clear for the Participant	The Wording Was Clear for the Participant
1	100% (51)	98% (50)	90.2% (46)
2	100% (51)	98% (50)	90.2% (46)
3	98% (50)	96.1% (49)	96.1% (49)
4	100% (51)	94.1% (48)	96.1% (49)
5	98% (50)	98% (50)	98% (50)
6	98% (50)	94.1% (48)	98% (50)
7	98% (50)	88.2% (45)	88.2% (45)
8	100% (51)	96.1% (49)	94.1% (48)
9	100% (51)	100% (51)	100% (51)
10	100% (51)	98% (50)	100% (51)
11	98% (50)	94.1% (48)	96.1% (49)
12	100% (51)	100% (51)	100% (51)
13	100% (51)	90.2% (46)	94.1% (48)
14	92.2% (47)	94.1% (48)	98% (50)
15	100% (51)	94.1% (48)	100% (51)
16	96.1% (49)	96.1% (49)	92.2% (47)
Overall (%)	98.6%	95.6%	95.7%

**Table 6 behavsci-12-00183-t006:** Means (SD) and correlations between participant characteristics and total sedentary time during weekdays and weekends (*n* = 51).

Variable	Classification	Total Sedentary Time (Hours/Day)					
		Weekdays		*p*-Value	Weekends		*p*-Value
		Mean (SD)	Correlation		Mean (SD)	Correlation	
Gender	Male	13.4 (4.0)	−0.029 *p* = 0.840	0.840	14.8 (3.8)	−0.051 *p* = 0.724	0.724
	Female	13.2 (3.3)			14.4 (4.1)		
Age category	Adolescents	14.1 (2.8)	−0.059 *p* = 0.680	0.218	14.5 (3.6)	−0.111 *p* = 0.438	0.360
	Young adult	13.6 (3.4)			16.3 (2.6)		
	Middle age	11.4 (3.5)			13.6 (3.8)		
	Older age	14.0 (4.3)			14.2 (4.9)		
BMI category	Underweight	17.0 (0.0)	−0.385 *p* = 0.005	0.046 *	17.3 (1.2)	−0.018 *p* = 0.903	0.532
	Normal weight	14.6 (3.0)			13.8 (4.1)		
	Overweight	12.9 (3.6)			14.9 (4.0)		
	Obesity	11.9 (3.9)			14.6 (3.9)		
Education	High school	13.9 (3.5)	−0.020 *p* = 0.888	0.144	14.4 (3.9)	0.018 *p* = 0.899	0.887
	College degree	11.9 (3.8)			15.0 (3.7)		
	Post graduate	14.3 (3.2)			14.5 (4.3)		
Work status	Not working	13.5 (4.2)	−0.099 *p* = 0.488	0.631	13.7 (4.9)	0.115 *p* = 0.423	0.602
	Working online	13.9 (2.8)			15.0 (3.5)		
	Working in-person	13.8 (3.8)			14.9 (3.5)		
Comorbidity	No	13.5 (3.2)	−0.041 *p* = 0.778	0.152	14.4 (4.1)	0.060 *p* = 0.673	0.627
	Yes	13.2 (4.1)			14.9 (3.8)		

* Multiple comparisons (Bonferroni tests) were not significant.

**Table 7 behavsci-12-00183-t007:** Correlation coefficients of total sedentary time with the questionnaire items during weekdays and weekends (*n* = 51).

Item (Question) Number	Correlation with Total Sedentary Time during Weekdays	Correlation with Total Sedentary Time during Weekends
1	−0.153 (*p* = 0.284)	-
2	−0.062 (*p* = 0.665)	-
3	0.576 (*p* < 0.001)	0.516 (*p* < 0.001)
4	0.473 (*p* < 0.001)	0.446 (*p* = 0.001)
5	0.245 (*p* = 0.083)	0.224 (*p* = 0.114)
6	0.195 (*p* = 0.171)	0.195 (*p* = 0.170)
7	0.124 (*p* = 0.386)	0.340 (*p* = 0.015)
8	0.443 (*p* = 0.001)	0.160 (*p* = 0.262)
9	0.353 (*p* = 0.011)	0.375 (*p* = 0.007)
10	0.206 (*p* = 0.146)	0.225 (*p* = 0.112)
11	0.198 (*p* = 0.165)	0.114 (*p* = 0.427)
12	0.259 (*p* = 0.066)	0.357 (*p* = 0.010)
13	0.293 (*p* = 0.037)	0.219 (*p* = 0.122)
14	0.179 (*p* = 0.208)	0.134 (*p* = 0.349)
15	0.263 (*p* = 0.096)	0.153 (*p* = 0.284)
16	−0.225 (*p* = 0.113)	0.134 (*p* = 0.349)

**Table 8 behavsci-12-00183-t008:** Percent variances explained by the components of the questionnaire for the weekdays and weekends items.

Component	Initial Eigenvalues			Rotation Sum of Squared Loading		
	Total	% Variance	Cumulative %	Total	% Variance	Cumulative %
**Weekdays**						
1	2.446	18.814	18.814	1.887	14.517	14.517
2	1.695	13.039	31.853	1.521	11.701	26.218
3	1.544	11.876	43.729	1.518	11.679	37.897
4	1.361	10.473	54.202	1.509	11.608	49.505
5	1.047	8.055	62.257	1.440	11.076	60.581
6	1.024	7.881	70.137	1.242	9.556	70.137
7	0.876	6.739	76.876			
8	0.711	5.470	82.346			
9	0.684	5.262	87.608			
10	0.488	3.757	91.364			
11	0.450	3.459	94.824			
12	0.411	3.165	97.989			
13	0.261	2.011	100.00			
**Weekends**						
1	2.242	17.244	17.244	1.940	14.927	14.927
2	1.849	14.224	31.468	1.762	13.557	28.484
3	1.516	11.663	43.131	1.604	12.339	40.823
4	1.257	9.673	52.804	1.356	10.431	51.254
5	1.142	8.782	61.586	1.343	10.331	61.586
6	0.993	7.637	69.223			
7	0.787	6.050	75.273			
8	0.764	5.875	81.148			
9	0.605	4.656	85.804			
10	0.569	4.381	90.185			
11	0.503	3.869	94.054			
12	0.416	3.201	97.254			
13	0.357	2.746	100.00			

**Table 9 behavsci-12-00183-t009:** Rotated component matrix and factor loadings for sedentary behaviors items in weekdays and weekends *.

Component	Weekdays		Weekends	
	Item (Question) Number	Coefficient	Item (Question) Number	Coefficient
1	10	0.814	4	0.716
	5	−0.666	13	0.689
	7	0.638	3	0.550
	15	0.499	15	0.451
2	9	0.782	10	0.766
	8	0.663	9	0.677
	-	-	11	0.618
3	3	0.860	12	0.749
	13	0.634	6	0.735
4	12	0.783	5	−0.669
	11	0.757	14	0.637
5	6	0.791	7	0.864
	14	0.655	8	−0.645
6	4	0.906	-	-

* Extracted method: principal component analysis. Rotation method: Varimax with Kaiser normalization.

## Data Availability

All data generated or analyzed during this study are included in this published article. Any additional data will be available from the corresponding author upon reasonable request.

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
