# Peer review of "An Arabic Sedentary Behaviors Questionnaire (ASBQ): Development, Content Validation, and Pre-Testing Findings"

_behavsci, 2022, doi:10.3390/bs12060183_

Round 1

Reviewer 1 Report

  1. The problem of research is unclear and cannot be treated as a dilemma for scientific consideration.
  2. The aim of the work is in fact to present a non-categorically scientific research tool (the subject of the research) and to deliberate on its accuracy. It is also not known whether the goal is to create a research tool, or to test it, or to collect the opinions of "experts"?
  3. The accuracy of the test (usefulness of a scientific tool) and its value assessed on a multi-point scale have been entrusted to "experts", which cannot be the essence of scientific research.
  4. The results are presented as duplicate (in writing and in tables), and their significance for solving the vague problem and the aim of the study is marginal.
  5. The presentation of many results not related to the purpose of the work and the statistical procedures carried out in connection with them are excessively complex or redundant.
  6. The discussion is not a confrontation with the literature on the subject, but rather a continuation of the description of own research, ending with general reflections, poorly related to the unclear purpose of the work.

Reviewer 2 Report

I thank you for the opportunity to read this work. The topic is interesting and certainly important for promoting health.

However, I do not think that the specificity relating to Arab culture, which motivates the construction of a new instrument, rather than the translation of existing ones, emerges clearly enough.

Furthermore, some themes are repeated in different parts of the paper. Rewriting some parts with more care to follow a linear logical thread would improve the work

10-11 It may be useful to add a connecting phrase

11 “… relate sedentary time to social, environmental, and health outcomes”

It may be a possible search, but it is not data evaluated by the instrument.

Introduction

Rewrite health vs. disease topics in more orderly sequence, not break them up in the text

80 It may be useful to add a connecting phrase

87- 94 Rewrite by rearranging the sequence of sentences. Possibly explain (89) why self-reports "are more strongly associated with health outcomes than objective measures".

126- 136 Better to provide further guidance. It is not clear what criteria were adopted for the selection. Why do only three of the 35 questionnaires (117) or 22 (124) are suitable? Why do there are characteristics of only two? What is "the latter instrument"? In the cited bibliography it is better to also include the articles in which the questionnaires are presented by the authors.

137- 138 After “extensive discussion” among research team members, it was noted that one of  the sedentary behavior instruments was too long.   Rewrite

161 “After defining the construct”.

            Is this a different definition from the one given at 28-29?

162  “ we generated the questionnaire item”

Perhaps it might be useful to explicitly link the construct with the items (perhaps a reference to table 2 could be included?).

171- 172 It is useful to explain why such a wide range was chosen (also indicate this, in the introduction, in the scope of the project)

Table 4 Body weight (kg) Mean (SD) 72.9 (15.3) Height (cm)  are not indicative in such a large age sample . I would remove them

Table 6 Check terms and values in bold

Uniform indication: p non p

519 bold

523 to wrap into the next line

In the discussion, it would be more interesting to highlight the specificity of the proposed questionnaire (apart from the language) also in relation to existing instruments. The indicated criteria (148- 160) are present in other questionnaires, why is the construction of the ASBQ important?

A description/interpretation of the 6 components of the questionnaire could be proposed.

In general, the discussion could be rewritten with a stricter logical thread.

Review the bibliography according to the journal standards

Author Response

Comments and Suggestions for Authors

 Reviewer’s comment:

However, I do not think that the specificity relating to Arab culture, which motivates the construction of a new instrument, rather than the translation of existing ones, emerges clearly enough.

Authors’ answer:

We studied almost all published SB questionnaires in English language and found them less suitable for our intents and purposes (see our criteria in the methods section). However, we include additional sentences at the end of the third paragraph stating that “To improve knowledge and utility of SB questionnaires and to reduce the gaps in assessing sedentary behaviors subjectively and their effects on health outcomes, it is essential that multiple characteristics of SB be assessed such as doing arts, crafts, and hobbies [27], while keeping the questionnaire as short and manageable as possible.” Our Arabic SB questionnaire provides overall as well as specific sedentary behavior estimates during a variety of sedentary activities covering leisure-time activities, work, transport, household seated activities, reading, chatting, and listening to religious verses or music. It also can be used for students, working peoples, unemployed, and retired persons as well as adolescents, and young-, middle-, and older-aged people.

Reviewer’s comment:

Furthermore, some themes are repeated in different parts of the paper. Rewriting some parts with more care to follow a linear logical thread would improve the work.

Authors’ answer:

We tried to edit the manuscript to improve the flow and reduce as much redundancy as possible.

Reviewer’s comment:

10-11 It may be useful to add a connecting phrase

Authors’ answer:

We edited line 10 in the abstract. Also, we added a connecting phrase in lines 10-11 in the introduction. 

Reviewer’s comment:

11 “… relate sedentary time to social, environmental, and health outcomes” It may be a possible search, but it is not data evaluated by the instrument.

Authors’ answer:

Thanks for this suggestion. However, some other reviewers suggested cutting down of the introduction section and we did. Adding one more paragraph on the social, environmental, and health outcomes will make it too long, besides that, as you have mentioned, it is not the focus of the current research.

Reviewer’s comment:

Introduction

Rewrite health vs. disease topics in more orderly sequence, not break them up in the text.

Authors’ answer:

W edited the introduction and cut down some sentences, thus, eliminating the fragmentation problems in paragraphs 2 and 3.

Reviewer’s comment:

80 It may be useful to add a connecting phrase

Authors’ answer:

We could not see a need to do anything in line 80. However, we did edited many paragraphs in the introduction.

Reviewer’s comment:

87- 94 Rewrite by rearranging the sequence of sentences. Possibly explain (89) why self-reports "are more strongly associated with health outcomes than objective measures".

Authors’ answer:

We added the following sentence explaining the reason for this difference:

“As the current evidence [7] indicates that screen time (especially TV viewing) obtained by SB questionnaires has a bigger impact on health compared with overall sedentary time that is attained by motion sensors.”

Reviewer’s comment:

126- 136 Better to provide further guidance. It is not clear what criteria were adopted for the selection. Why do only three of the 35 questionnaires (117) or 22 (124) are suitable? Why do there are characteristics of only two? What is "the latter instrument"? In the cited bibliography it is better to also include the articles in which the questionnaires are presented by the authors.

Authors’ answer:

We added the criteria used and the third article that was discussed by the team has been added as well.   

Reviewer’s comment:

137- 138 After “extensive discussion” among research team members, it was noted that one of the sedentary behavior instruments was too long.   Rewrite

Authors’ answer:

We rewrite this statement.

Reviewer’s comment:

161 “After defining the construct”.

            Is this a different definition from the one given at 28-29?

Authors’ answer:

It is the same. We rephrase that statement.

Reviewer’s comment:

162  “ we generated the questionnaire item”

Authors’ answer:

Thanks. We correct the wording.

Reviewer’s comment:

Perhaps it might be useful to explicitly link the construct with the items (perhaps a reference to table 2 could be included?).

Authors’ answer:

Thanks for the note. We added the references.

Reviewer’s comment:

171- 172 It is useful to explain why such a wide range was chosen (also indicate this, in the introduction, in the scope of the project)

Authors’ answer:

This is one of our criteria, as to be applicable to a wide range of ages. It was missing and was added now in the revised version of the manuscript. This was also added at the end of the introduction.

Reviewer’s comment:

Table 4 Body weight (kg) Mean (SD) 72.9 (15.3) Height (cm) are not indicative in such a large age sample . I would remove them

Authors’ answer:

We agree with reviewer that they are not exactly indicative of the actual means due the wider age range. However the reader may be interested in generally knowing the means and SD’s of the weight and height of those participating in the pre-testing phase.

Reviewer’s comment:

Table 6 Check terms and values in bold

Uniform indication: non p

Authors’ answer:

Thanks. We remove the bold and fixed the p values.

Reviewer’s comment:

519 bold

Authors’ answer:

Bold fonts was eliminated

Reviewer’s comment:

523 to wrap into the next line

Authors’ answer:

It was fixed.

Reviewer’s comment:

In the discussion, it would be more interesting to highlight the specificity of the proposed questionnaire (apart from the language) also in relation to existing instruments. The indicated criteria (148- 160) are present in other questionnaires, why is the construction of the ASBQ important?

Authors’ answer:

We added some additional sentences at the end of first paragraph showing the features of the Arabic SBQ.

Reviewer’s comment:

A description/interpretation of the 6 components of the questionnaire could be proposed.

Authors’ answer:

The description of the components can be easily derived from appendix 3. However, we added few sentences elaborating on this aspect.

 Reviewer’s comment:

In general, the discussion could be rewritten with a stricter logical thread.

Authors’ answer:

The discussion was edited in the revised manuscript.

Reviewer’s comment:

Review the bibliography according to the journal standards

Authors’ answer:

We went over the reference format and fixed it.

Reviewer 3 Report

Thank you for the opportunity to review this interesting manuscript. It is about the development of a questionnaire to detect sedentary behaviors for Arabic people. I found the manuscript interesting, well presented and structured. It is specific for a Population but it is useful for future studies. I suggest its publication after minor revisions.

The introduction is clear and well structured. I have only one suggestion for line 68-80: according my opinion, this part is a repetition of the first part of the introduction.

In the methods, if a systematic review was performed, please, provide the guidelines followed such as PRISMA. Otherwise change it in “review”.

Results and discussion are well presented and structured.

To be more specific:

The applicability of the researchers questionaire is specific for a population limiting its use and interest of the community. A second important limitation of the research is the inclusion of internal expert, importantly limiting the quality of the research.  Furthermore, the scale has not been properly validated. The manuscript has no strength points. The first part of the discussion (lines 388-410) requires references. I have no other comments for this manuscript in this first stage of revision.

Author Response

Comments and Suggestions for Authors

Reviewer’s comment:

The introduction is clear and well structured. I have only one suggestion for line 68-80: according my opinion, this part is a repetition of the first part of the introduction.

Authors’ answer:

Thanks for this comment. The paragraph from line 68-80 talks about the effect of breaking sitting time with physical activity, which appears to eliminate the increased risk of death associated with high sitting time. On the other hand, the second paragraph discusses how SB is considered an important risk factor for mortality and several chronic diseases.

However, we edited the introduction section and reduced some of redundant phrases.

Reviewer’s comment:

In the methods, if a systematic review was performed, please, provide the guidelines followed such as PRISMA. Otherwise change it in “review”.

Authors’ answer:

We stated in the second paragraph under the subtitle “Arabic SB questionnaire (ASBQ) development” that “we have first conducted an extensive literature search”, so we did not use the word “systematic”.

Reviewer’s comment:

Results and discussion are well presented and structured.

Authors’ answer:

Thanks.

Reviewer’s comment:

To be more specific:

The applicability of the researchers questionaire is specific for a population limiting its use and interest of the community. A second important limitation of the research is the inclusion of internal expert, importantly limiting the quality of the research.  Furthermore, the scale has not been properly validated. The manuscript has no strength points. The first part of the discussion (lines 388-410) requires references. I have no other comments for this manuscript in this first stage of revision.

Authors’ answer:

The current manuscript reports on the first phase of the ASBQ; that is developing the questions within the construct of SB, testing the content validity, and present the pre-testing findings. The next steps of course will involve undergoing further testing of psychometric properties, including internal consistency, test-retest reliability, and criterion validity against an objective measure of SB.

As to the applicability and generalizability of the questionnaire, we used a wide range age groups, including adolescent, and young-, middle-, and older-age groups, of both males and females as well as included all probable sedentary activities from variety of domains. Further testing of instrument will provide more insight into further suitability of this instrument.

For testing the content validity of any new subjective instrument, experts opinion and agreement is the most common procedures (kindly, see references 53-56) and this what we did.

As to citing references for the first paragraph of the discussion, this is an introductory paragraph representing a summary of the most important findings of this study. For the second paragraph of the discussion, we included a couple of references (references 59 and 61).

Round 2

Reviewer 1 Report

The article still does not meet the methodological criteria for scientific works.

All comments made in the previous review remain valid.